# Distinguishing local isomorphism classes in quasicrystals by high-order harmonic spectroscopy

Jia-Qi Liu ⓘ & Xue-Bin Bian ⓘ ✉

Electron diffraction spectroscopy is a fundamental tool for investigating quasicrystal structures, which unveils the quasiperiodic long-range order. Nevertheless, it falls short in effectively distinguishing separate local isomorphism classes. This is a long outstanding problem. Here, we study the high-order harmonic generation in two-dimensional generalized Penrose quasicrystals to optically resolve different local isomorphism classes. The results reveal that: (i) harmonic spectra from different parts of a quasicrystal are identical, even though their atomic arrangements vary significantly. (ii) The harmonic yields of diverse local isomorphism classes exhibit variations, providing a way to distinguish local isomorphism classes. (iii) The rotational symmetry of harmonic yield can serve as a characteristic of quasicrystal harmonics and is consistent with the orientation order. Our results not only pave the way for confirming the experimental reproducibility of quasicrystal harmonics and identifying quasicrystal local isomorphism classes, but also shed light on comprehending electron dynamics influenced by the vertex environments.

Choices of the unit cell and packing methods in crystals exhibit a strict one-to-one correspondence to form a single undefeated periodic structure. Very different from crystals, two or more unit cells can be assembled into an infinite number of various quasicrystals, which can be further subdivided into local isomorphism classes (LICs)[1,2]. Two quasicrystals are local isomorphic if and only if every local arrangement with any finite size in one is also present in the other. Since the discovery of quasicrystals[3], electron diffraction spectroscopy (EDS) has been a gold standard for extracting the symmetry and finding new quasicrystal materials. Two quasicrystals have identical diffraction patterns if and only if they are local isomorphic[1]. Diffraction peaks of distinct LICs have the same locations but slightly different intensities. However, we will see that, as illustrated below in the two-dimensional (2D) generalized Penrose quasicrystals, the subtle EDS disparity between diverse LICs limits the discrimination. Since different LICs exhibit significant disparities in physical properties, including free energy, entropy[1,2], electron

localization[4], and superuniformity[5,6], an alternative approach is requisite to distinguish various quasicrystals.

High-order harmonic generation (HHG) from gases is one of the classical products of nonlinear laser-matter interactions[7,8]. It is important in compressing laser pulse width[9] and obtaining attosecond light sources[10]. Crystals[11–16] and liquids[17,18] have recently garnered significant interest as sources of HHG. Decoding HHG to retrieve the microstructure and electron properties of matter is one of the crucial topics in ultrafast optics. Significant progress has been made in atomic orbital imaging[19–21], chiral molecule inspection[22–24], crystal band reconstruction[15,25], valence electron imaging[26], and topological phase transition detection[27,28]. Compared to EDS, lasers afford more control freedoms, such as polarization, waveform, ellipticity, and so on, which offer distinctive avenues for imaging quasicrystal structures. Nonetheless, research in this field is currently rare.

Quasicrystals, lying between periodic crystals and disordered amorphous materials (like liquids), are renowned for their unique

Wuhan Institute of Physics and Mathematics, Innovation Academy for Precision Measurement Science and Technology, Chinese Academy of Sciences, Wuhan 430071, China. ✉e-mail: xuebin.bian@wipm.ac.cn

order, which may lead to novel harmonic generation mechanisms and a new type of HHG source[29–31]. On the other hand, the orientation-dependent HHG in quasicrystals can be expected to reveal their forbidden rotational symmetries, serving as a fingerprint for experimental identification. Furthermore, HHGs capture ultrafast electron dynamics and offer a potential all-optical method to explore both structures and electron properties of quasicrystals. As we all know, imperfect symmetry and disorder are always present in real solids, such as glass, ceramics, polycrystals, and crystals at room temperature. Research on these aspects is very important for the development of attosecond science, but most of them are skipped due to the complexity of the system. From this perspective, the study of quasicrystal HHG is very relevant for understanding the harmonic generation mechanism in complex systems, which allows us to study complex electron dynamic processes in an ideal ordered system.

Demonstrating distinctive rotational symmetry in EDS, 2D quasicrystals have simpler spatial structures compared to three-dimensional settings and have real physical counterparts, like 30° twisted bilayer graphene[32] and the alloys with 8[33], 10[34], and 12-fold[35] symmetries. Therefore, studying HHG in 2D quasicrystals is both more feasible and essential. Recently, the topological properties[36–38], and cold atom simulation[39–41] of 2D quasicrystals have also received extensive attention. As the classical paradigm and pioneer of quasicrystals, the static electron properties of 2D Penrose tiling[42] have been widely studied in the density of states[43,44] and the localization of wave functions[45]. To generate Penrose tiling, various methods have been developed, including matching rules[42,46], self-similarity transformations[45], the generalized dual method (GDM)[1,2], and superspace projection[47–52]. Notably, including Penrose tilings, GDM and five-dimensional superspace projection can generate various quasiperiodic tilings, which are collectively termed generalized Penrose tilings[49–52]. These tilings share the same fat and thin rhombic unit cells but can belong to different LICs.

In this work, we take 2D generalized Penrose quasicrystals as an example to distinguish different LICs by HHG. We observed distinct differences in HHG yield and orientation dependence among LICs, allowing for their differentiation. These variations arise from differences in vertex environments (VEs), linked to the system's hyperuniformity. Furthermore, the orientation-dependent HHG retains the crystallography-forbidden tenfold rotational symmetry, characteristic of quasicrystal harmonic radiation. Atomic units (a.u.) are used throughout unless stated otherwise.

## Results
### Generalized Penrose quasicrystal model
Using GDM[1,2] based on 5-grids, we construct a series of 2D quasiperiodic tilings. The star vectors $\{\mathbf{e}_i\}_{i=0,1,2,3,4}$ in Fig. 1a represent the orientation order, with $\mathbf{e}_i = (\cos(2\pi/5i + \delta_0), \sin(2\pi/5i + \delta_0))$, and $\delta_0 = 2\pi/30$. Perpendicular to each $\mathbf{e}_i$, one can construct a periodic 5-grid (the gray solid lines), which determines the quasiperiodic order. As exemplified by the 0th grid lines marked with red dashed lines, the lines are numbered in each grid to denote the open regions split by the 5-grid into five integers $(k_0, k_1, \cdots, k_4)^{1,2}$. The tiling and rhombuses vertices $\{\mathbf{t}\}$ can be obtained by the dual transformation, $\mathbf{t} = \sum_{i=0}^{4} k_i \mathbf{e}_i$. The algebraic distance between the 0th line of $i$th grid and the origin is defined as the parameter $\gamma_i$, which determines the grid plane and its dual. Socolar et al.[2] proved a critical theorem: If and only if $\{\gamma_i\}$ and $\{\gamma_i'\}$ of two 5-grids are satisfied, $\sum_{i=0}^{4}\gamma_i - \sum_{i=0}^{4}\gamma_i' = m, m \in \mathbb{Z}$, their

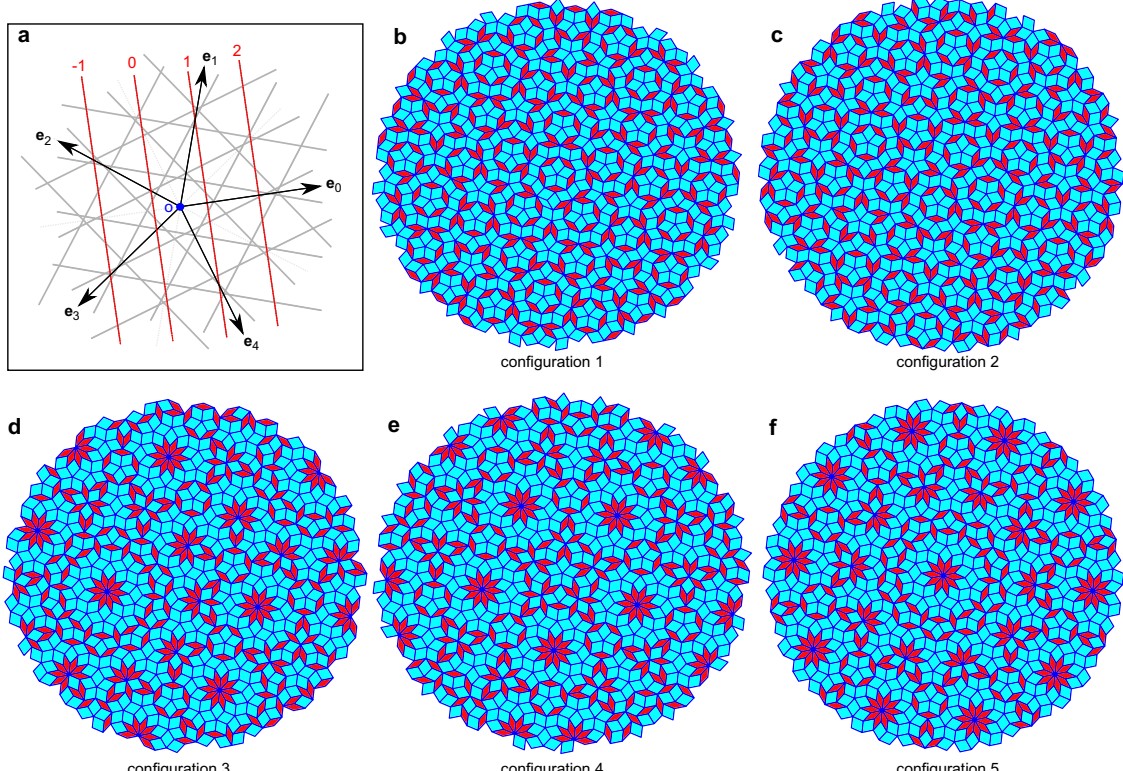

**Fig. 1 | Several 2D generalized Penrose quasicrystal configurations obtained by GDM. a** Schematic diagram of the periodic 5-grid in the GDM. Black arrows and gray grids represent the star vectors and the 5-grid, respectively, with the 0th grid line indicated by red dashed lines. **b**–**f** respectively denote the configurations 1–5 obtained by the GDM, including fat and thin rhombuses represented in light blue and red. The number of atoms $N_{\mathrm{atom}}$ (or vertices) in each configuration is successively 991, 981, 975, 991, and 981. Configurations 1 and 2, belonging to LIC1 (the Penrose LIC), have grid parameters $\{\gamma_i^1 = -0.2\}$ and $\{\gamma_0^2 = -0.48, \gamma_1^2 = -0.39, \gamma_2^2 = -0.50, \gamma_3^2 = -0.29, \gamma_4^2 = -0.34\}$, respectively. Configurations 3–5 belong to LIC2 for $\lfloor \sum_{i=0}^{4}\gamma_i \rfloor - \sum_{i=0}^{4}\gamma_i = 0.5$ ($\lfloor \rfloor$ is the "floor" operator), and the grid parameters are $\{\gamma_0^3 = -0.15, \gamma_1^3 = -0.22, \gamma_2^3 = -0.23, \gamma_3^3 = -0.42, \gamma_4^3 = -0.48\}$, $\{\gamma_i^4 = -0.3\}$, and $\{\gamma_i^5 = -0.5\}$, respectively. If $\gamma_0 = \gamma_1 = \gamma_2 = \gamma_3 = \gamma_4$, the dual tiling will have a symmetric center. Source data are provided as a Source Data file.

dual tilings belong to the same LIC. Hence, the parameter $\gamma$, the fractional part of $\sum_{i=0}^{4} \gamma_i$, dictates the LIC of configurations, and its absolute value can be consistently mapped to the interval [0, 0.5]. In particular, when $\gamma = 0$, the tiling belongs to the Penrose LIC and can be obtained by the "matching rule"[42,46]. Varying $\{\gamma_i\}$ and enlarging the rhombuses side length $l$ to 8 a.u., we obtain a series of tilings, exhibited in Fig. 1. Configurations 1 and 2 belong to LIC1 ($\gamma = 0$), and configurations 3–5 are included in LIC2 ($\gamma = 0.5$). The localized Gaussian potentials are adopted to describe the atomic potential located on the rhombus vertices[18]. Configurations' energy spectrum and eigenstates can be obtained by diagonalizing the field-free Hamiltonian. Taking configuration 1 as an example, we recognize the local, critical, and extended eigenstates, which are consistent with the tight-binding central model[45]. The light localization in photonic quasicrystal[4] is also reproduced by the electron states of configuration 3. This evidence confirms the rationality of our model (see Supplementary Note 1 and Supplementary Figs. 1 and 2). From the energy spectrum (Supplementary Fig. 1), our models have a bandgap of about 5 eV, similar to typical insulators (>2 eV). We will see that the interband harmonics play a major role in distinguishing LICs. With suitable laser conditions, our results should be applicable to other gapped quasicrystal models.

## Electron diffraction spectroscopy

Since our model involves vertex-decoration-only structures with a single atom type at the vertices, the EDS (or structure factor) can be simply obtained via Fourier transform of the atomic positions (see Supplementary Note 2 for detail), without requiring complex corrections for atom types, distributions, or disorder[53]. EDSs of different configurations in the two kinds of LICs have been calculated (see Supplementary Note 2 and Supplementary Fig. 3). Although our models possess finite size, we can still arrive at the same conclusion as elucidated by Levine et al.[1]. The local isomorphic configurations possess nearly identical EDS, while varying LICs exhibit subtle variations in diffraction peak intensities, underscoring that EDS is ineffective in discriminating different LICs.

## Simulation of electron dynamics in quasicrystals

The laser-quasicrystal interaction is simulated by solving the time-dependent Schrödinger equation (TDSE). Similar to the time-dependent density functional theory with frozen Kohn–Sham potentials[27,54], $N_{atom}$ valence states below zero energy are selected as initial states and evolve independently. The HHG spectra are obtained by the Fourier transform of the laser-induced currents, and the yield of the $n$th order harmonic is $I_n$.

## Influence of laser spot position

As illustrated in Fig. 2a, unlike periodic crystals, spot irradiation at different positions (A or B) on the quasicrystal may encounter diverse atomic structures. In two independent experiments, the lasers always cover distinct atomic patterns. Investigating the impact of spot position variations on quasicrystal HHG will aid in confirming experimental reproducibility, which can be achieved by comparing harmonic responses among local isomorphic configurations, according to the definition of LIC. As depicted in the schematic diagram in Fig. 2b, we investigate the scenario where a linearly polarized laser illuminates a 2D quasicrystal vertically. First, we fix the polarization angle $\theta$ at 0° and configure the laser with wavelength $\lambda = 2.6\,\mu m$ and total duration $L_t = 10T$ ($T$ is the optical period). Short pulse duration and low photon energy mean a small absorbing cross-section, enabling higher laser intensities for nonperturbative effects without damaging the target. In Fig. 2c, d, HHGs and the phases (see Supplementary Fig. 4) of configurations in one LIC with apparent structure differences are almost identical. In fact, this conclusion is robust for the laser's intensity, wavelength, and polarization angle (see Supplementary Note 3 and Supplementary Figs. 5–7). In a circular polarization scenario, when the

number of atoms reaches about 6100, the harmonic selection rule[55–57] of each configuration aligns with $10k \pm 1$ ($k \in \mathbb{N}$) (see Supplementary Fig. 8), consistent with quasicrystal orientational order (determined by the star vector in Fig. 1a) and reciprocal lattice symmetry. Consequently, we predict identical harmonic radiation in actual experiments when the laser irradiates different positions on the quasicrystal, irrespective of linear, circular polarization, or any ellipticity in between. The unexpected insensitivity of quasicrystal harmonics to laser irradiation position can be clarified by considering the electron motion environment and the concept of LICs. In quasicrystals, HHG occurs as electrons initially localized at vertices are driven by a laser field in a finite region and scatter with atoms. Different initial vertices lead to distinct finite regions with varying atomic arrangements, which are shared among local isomorphic configurations. As noted, different configurations within the same LIC should produce nearly identical harmonic emissions. Focusing the laser on different parts of the quasicrystals reveals distinct atomic patterns, yet these irradiated regions belong to the same LIC, sharing the same electron motion environments and harmonic spectra. This will be elaborated upon in the following parts.

## HHG of different LICs

Now, let's revisit HHG's role in distinguishing LICs, with a primary focus on linear polarization driving lasers. To drive electrons far enough in the quasicrystals, we used an intensity of $I_0 = 3 \times 10^{13}$ W cm$^{-2}$. Similar laser intensity can be found in ref. 58. In Fig. 2e, we compared the HHG of configurations 1 and 3 from two different LICs and found a difference in yield: The 11th–19th order (the plateau region) harmonic yield of configuration 1 is higher than configuration 3, whose stability has been verified for $I_0 > 2 \times 10^{13}$ W cm$^{-2}$ and $\lambda > 1800$ nm (see Supplementary Figs. 5 and 6). We conclude that different LICs exhibit obvious variations in harmonic yield. Therefore, the HHG provides an alternative to determine whether two configurations belong to the same LIC. In the following parts, we will observe that variations in harmonic yield among different LICs correlate with the diversity of VEs. This affects the coherence of electron motion and harmonic radiation, characterizable through superuniformity.

Furthermore, we compare the orientation-dependent harmonic yields of diverse LICs. Figure 3a–f shows the orientation-dependent yield of the 11th to 21st-order harmonics for configuration 1 and configuration 3. Firstly, we find that the $\theta$-dependent harmonic spectra of LIC1 and LIC2 exhibit distinct intensity differences, enabling the identification of different LICs. As shown in Fig. 3a–c, the 11th, 13th, and 15th order harmonics of configuration 1 are notably stronger than configuration 3 across all orientations. In Fig. 3d and e, as polarization direction varies, the two non-local isomorphic configurations alternate in dominating the 17th and 19th-order harmonic radiation intensities. In Fig. 3f, configuration 3's 21st harmonic intensity exceeds that of configuration 1 for all polarization directions. Additional comparisons of orientation-dependent harmonic yields in various configurations are provided (see Supplementary Note 4 and Supplementary Figs. 9 and 10), further validating the effectiveness of $\theta$-dependent HHG in distinguishing LICs. We also compared the lower-order (e.g., 3rd-order) harmonics below the plateau region and found minimal differences. Thus, lower-order harmonics (in the perturbative regime) are not suitable for distinguishing LICs.

Secondly, let's delve into the symmetry of orientation-dependent HHG. As depicted in Fig. 3, the $\theta$-dependent harmonic yield of configurations 1 and 3 approximately exhibits 10-fold rotational symmetry, despite the absence of $C_{10}$ rotational symmetry in these configurations. This counterintuitive phenomenon can also be understood via the definition of the LIC and the finite electron motion range. Configurations 3 and 5 belong to the same LIC2, with the latter exhibiting 10-fold rotational symmetry (see Fig. 1). According to the LIC definition, in sufficiently large systems, any finite-size atomic

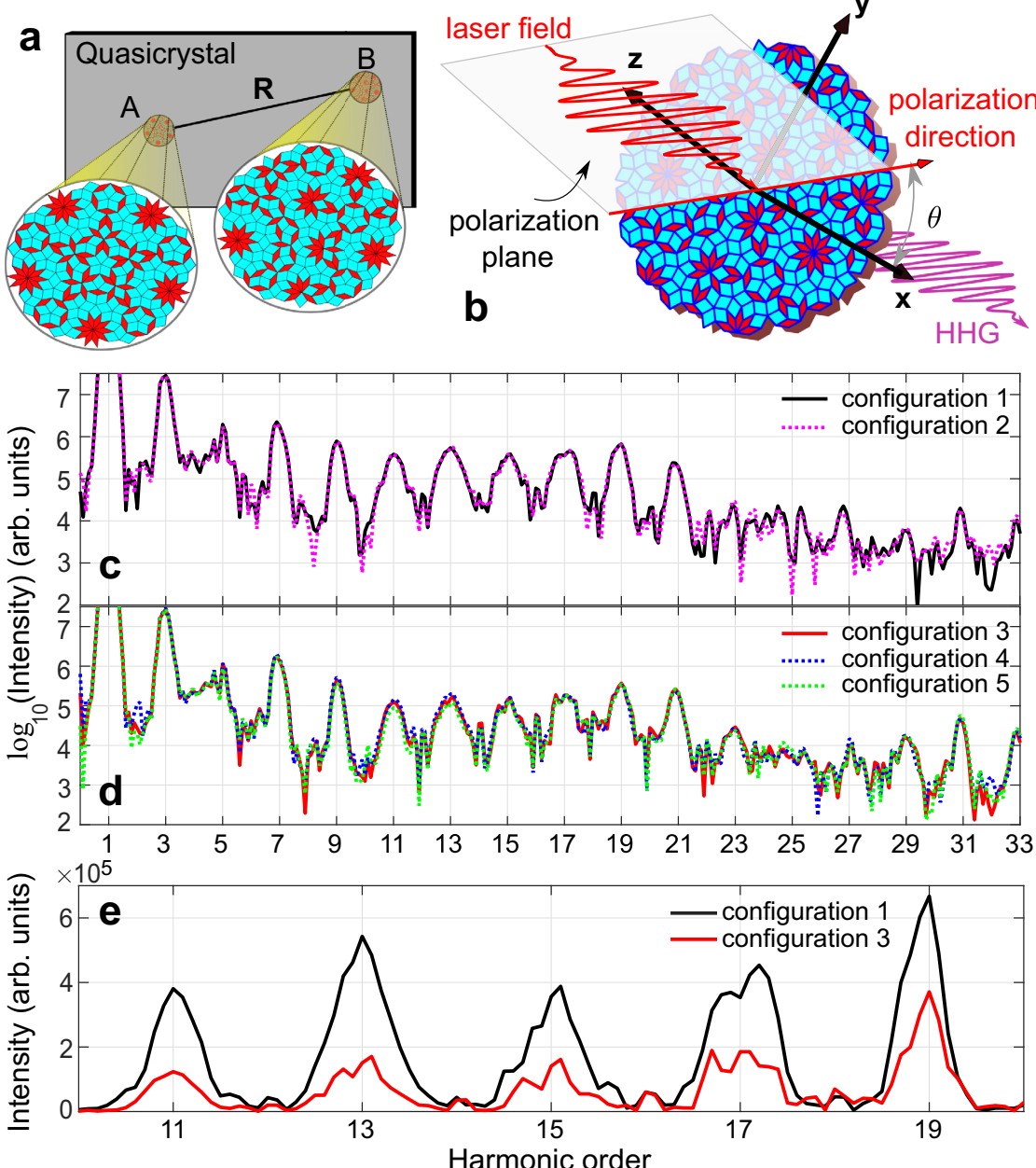

**Fig. 2 | Harmonic spectra comparison across different configurations.**
**a** Schematic diagram of laser irradiation at various positions on the quasicrystal.
**b** Schematic diagram of the interaction between generalized Penrose quasicrystals and linearly polarized laser field. The HHG spectra of configurations in (**c**) LIC1 and (**d**) LIC2, driven by a linearly polarized laser with an intensity of $I_0 = 3 \times 10^{13}$ W cm$^{-2}$, and sin$^2$ envelope. **e** Comparing 11th to 19th order harmonic spectra from configurations 1 and 3 of different LICs in linear coordinates. The harmonic spectra of configurations 1 to 5 are shown as solid or dashed lines in black, magenta, red, blue, and green, respectively. Source data are provided as a Source Data file.

arrangement in configuration 5 can be found in configuration 3. Under laser driving, the finite atomic arrangement experienced by an electron initially located at a vertice is shared by configurations 1 and 5, emitting identical harmonics. Hence, in sufficiently large systems, the orientation-dependent HHG of configuration 3 should align with configuration 5, exhibiting $C_{10}$ symmetry. Relative to configuration 1, we can identify another local isomorphic configuration $\bar{1}$ (with grid parameter $\{\gamma_i^{\bar{1}} = -0.8\}$) whose atomic structure is equivalent to rotating configuration 1 by 180° along the symmetry axis. Hence, any finite region in configuration 1 always appears in the configuration $\bar{1}$ in a rotated form by 180°. As configurations 1 and $\bar{1}$ are local isomorphic, any finite region of configuration $\bar{1}$ is always present in configuration 1. Thus, in sufficiently large configuration 1, there is always a pair of

electrons whose finite motion regions rotated 180° relative to each other, resulting in a $C_{10}$ symmetry for the orientation-dependent HHG of configuration 1 with $C_5$ symmetry. In our simulations, finite-size effects can induce subtle fluctuations in the harmonic spectra intensities of distinct configurations within an LIC, which leads to a slight deviation in the symmetry of the orientation-dependent HHG from $C_{10}$ and can be mitigated by expanding the system. Therefore, we conclude that for sufficiently large systems (such as the actual laser spot size), the $\theta$-dependent harmonic spectra of LIC1 and LIC2 will tend towards $C_{10}$, which is required by the quasicrystal's orientation order (determined via star vectors) and conforms to the EDS's symmetry. Another key element is that the orientation-dependent electron excitation probability and order-dependent electron motion trajectory

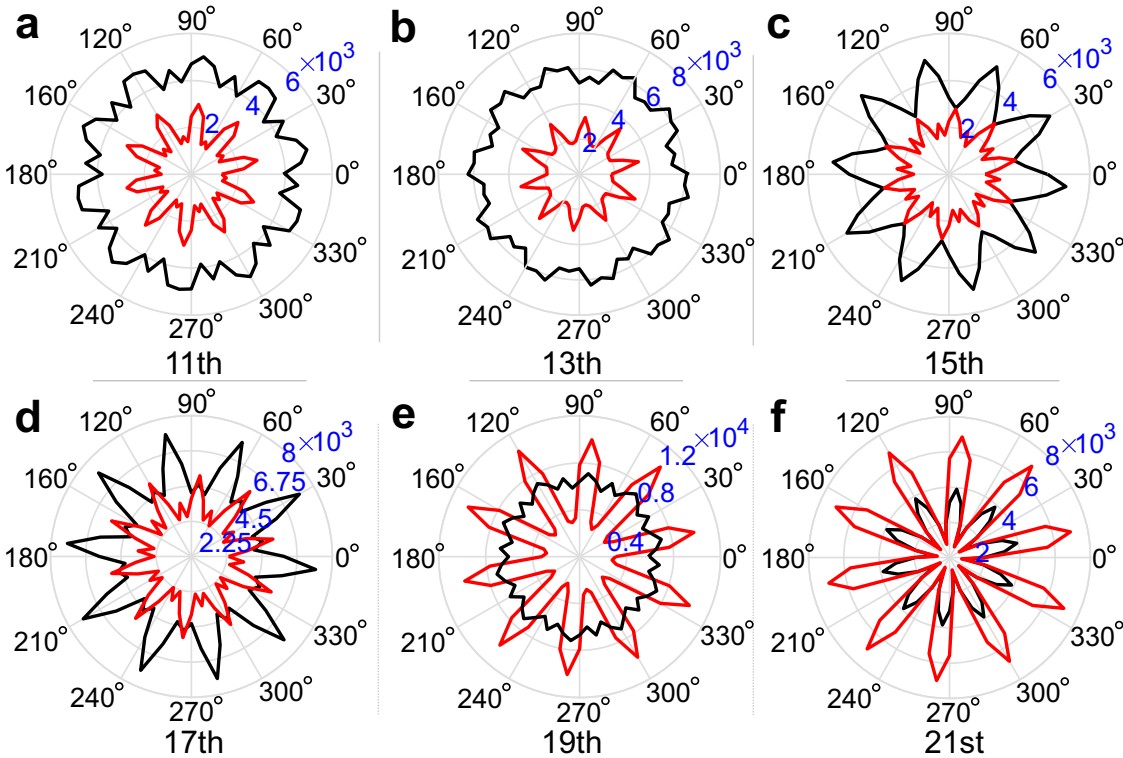

**Fig. 3 | Orientation-dependent harmonic yield. a–f** The $\theta$-dependent harmonic yield $I_n$ of the 11th–21st order in configuration 1 (LIC1, the solid black line) and configuration 3 (LIC2, the solid red line), with the radial representing harmonic yield (in arb. units). Source data are provided as a Source Data file.

together lead to different $\theta$-dependent shapes of harmonic yield across different orders. It's noteworthy that in our discussion, all configurations possess identical star vectors, a prerequisite for distinguishing different LICs. As depicted in Fig. 3, certain harmonics (e.g., 21st order) exhibit high yields in the star vectors and their opposite directions, offering a means to determine orientation order experimentally. As shown in the next two sections, differences in harmonic yield between LICs stem from the diversity of VEs, whose spatial distribution is governed by orientational and quasiperiodic order (modulated by $\gamma$), explaining the distinct HHG angular dependencies across LICs.

### Real-space electron motion perspective

It is known that in semiconductors and insulators, interband and intraband transitions play an important role in HHG[59–61]. Interband harmonics are often described as follows. The excited electron and hole wave packets are driven by the laser field to travel, and then they recombine at specific moments, emitting high-energy photons with gap energy. However, without the shallow-lattice approximation[41], our models' finite size and electron localization limit the application of energy bands in $k$ space. We can always sort the eigenvalues and divide the energy spectrum into the valence band (below zero) and conduction band (above zero) in the energy space (see Supplementary Fig. 1). By projecting the time-dependent wave function onto time-independent eigenstates[62], we distinguish intraband and interband harmonics in energy space. We find that the HHG plateau region, relevant for differentiating LICs, is mainly dominated by interband mechanisms (see Supplementary Note 5 and Supplementary Figs. 11 and 12), which are associated with the long-range motion of electrons[61]. So, we qualitatively analyze the relationship between harmonic radiation and LICs from the view of the excursion-electron dynamic environment. Taking the partial enlargement of valence electron density of configuration 1 in Fig. 4a as an example, the initial

electrons are mainly distributed at the vertices, and the VE[4,5,52] composed of rhombus tiles around a vertex will affect the dynamic process and harmonic radiation. Driven by a laser field, an electron from a vertex undergoes a complex motion trajectory and emits harmonic radiation through scattering with atoms on the vertices. The intricate multiple-scattering processes involving numerous atoms are challenging to track accurately. If electrons from different vertices observe different vertex patterns, the emitted harmonic radiations will exhibit suppressed coherence. According to the definition of LIC, a finite atomic arrangement centered around one vertex is consistently found in another configuration. So, as illustrated in Fig. 4b, a moiré pattern and the overlap regions (the light blue areas) can be generated by appropriately translating two isomorphic tilings sharing the same VEs. The overlap regions cover the vast majority of vertices in the two tilings, and the electrons in overlap regions undergo uniform scattering processes, which makes the almost uniform harmonic radiation. Thus, quasicrystal HHG remains unaffected by laser irradiation position, holding true across different laser conditions, including intensity, wavelength, orientation, and polarization.

On the other hand, the mismatch between the two patterns in Fig. 4b indeed corresponds to a phason flip for VEs Q or D[53] (see Fig. 5a and Supplementary Fig. 13). Phason flips between local isomorphic configurations affect only specific patches, not all Q and D. This contrasts with random phason flips applied to each patch, which introduce a tiling disorder[53]. The perspective on phason flips is enlightening. Perhaps we could relate HHG with the problem of phasonic disorder. Here, we try to investigate the impact of the phasonic disorder on harmonic yield in configuration 1 (see Supplementary Note 6). We found that as the disorder increases (controlled by flip iterations), the yields of some orders in the plateau region initially decrease and then recover, which can be qualitatively explained by the types and proportions of VEs (see Supplementary Fig. 14). A more detailed analysis of phasonic disorder's effects on HHG will be presented in future work.

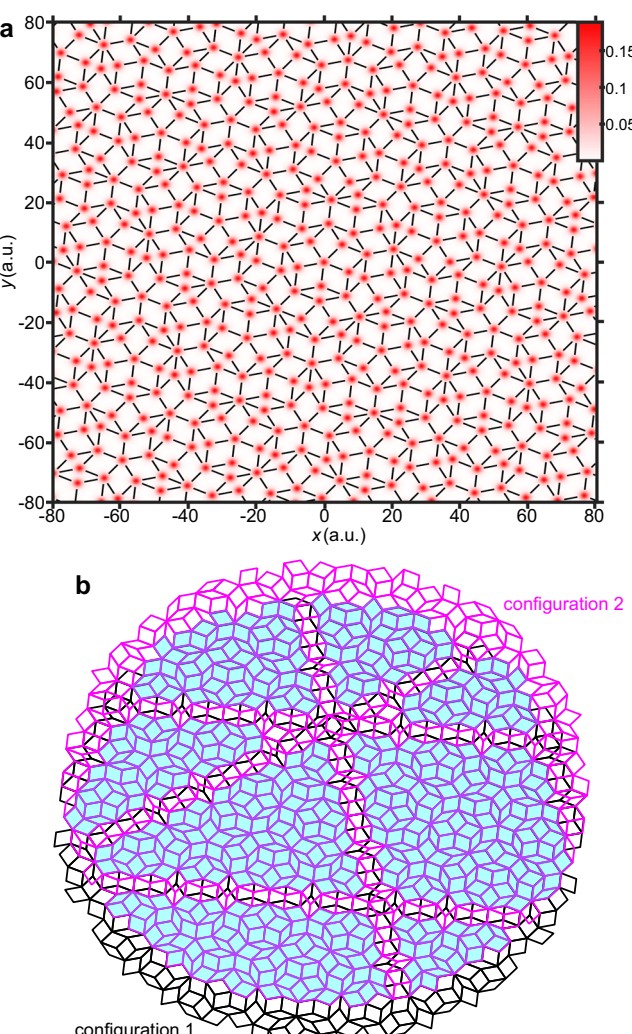

**Fig. 4 | Localized electron density and moiré patterns. a** Partial enlargement of the spatial distribution of electron density in configuration 1. **b** The moiré pattern built by local isomorphic configurations 1 (the black tiling) and 2 (the magenta tiling) under appropriate translation. The light blue region indicates the overlap between the two configurations. Source data are provided as a Source Data file.

## VEs and hyperuniformity

In this part, we aim to uncover the reasons behind the differences in harmonic intensity among various LICs. In generalized Penrose quasicrystals, fat and thin rhombi collectively create 16 distinct VEs[4,5,52], as illustrated in Fig. 5a. It is worth noting that within these VEs, M (defined by two fat and two thin rhombuses) consists of two inequivalent forms that are mirror images and cannot be transformed into each other by translation or rotation in the 2D plane. The two patterns of M always appear in pairs in the discussed configurations and contribute to the harmonic radiation simultaneously. In the following discussion, we will refer to both forms collectively as M. The frequency $F$[4,52], defined as the ratio of different VEs to the total vertices, varies with changes in LICs or parameter $\gamma$, as shown in Fig. 5b and c The Penrose LIC (LIC1) with $\gamma = 0$ is distinguished for having a fewest types of VEs, potentially explaining the higher harmonic yield observed in Fig. 2e. Taking the endpoint $p$ of the VE Q in Fig. 5a as an example, different LICs restrict the possible evolutions of $p$. In LIC1, $p$ has six choices (Q, K, J, S, T, V), while in LIC2, there are ten choices (Q, K, M, J, R, S, U, W, Y, Z). It shows that in different LICs, the VEs diversity at scattering point $p$ is different when electrons originate from Q. The analysis also applies to D or

other VEs. In fact, all VEs in the system are interconnected through complex electron scattering and collectively contribute to harmonic radiation. Thus, greater VE diversity and a more uniform $F$ distribution will lead to lower motion environment similarity for electrons originating from the same VEs but different positions, which reduces the HHG coherence and yield (as we will see in Fig. 5e). Using Voronoi cells as a metric, Lin et al.[5] emphasized that the local distribution of VEs governs quasicrystals' hyperuniformity, characterized by the suppression of long-wavelength density fluctuations. We expect that the variation of the hyperuniformity[5,6] will manifest in the harmonic spectrum of different LICs.

Let's discuss hyperuniformity briefly: considering quasicrystal atoms as a point pattern in $d$-dimensional Euclidean space $\mathbb{R}^d$, the number of points within a spherical window centered at $\mathbf{x}_i$ with radius $R$ is denoted as $N(R; \mathbf{x}_i)$, a random variable about $\mathbf{x}_i$ with variance $\sigma^2(R)$. If $\sigma^2(R)$ grows slower than the window volume, i.e., $\sigma^2(R) \sim R^\alpha, \alpha < d$, the system is hyperuniform. In the generalized Penrose quasicrystal ($\alpha = 1$), $\sigma^2(R)$ is expressed as $\sigma^2(R) \sim \Lambda(R)R + o(R)$, and $o(R)$ denotes terms of lower order than $R$[5]. Following ref. 5, we use the cumulative moving average $\bar{\Lambda}(R) = 1/R \int_0^R \Lambda(R')dR'$ to eliminate small-scale variations and take the asymptotic average $\bar{\Lambda}(\infty) = \lim_{R \to \infty} \bar{\Lambda}(R)$ to describe the hyperuniformity. In calculations, quasicrystal atoms ($N_{atom} \approx 980$) concentrate within a circle (radius $R_0 = 16l$) centered at the origin. $M = 10,000$ sampling window centers $\mathbf{x}_i$ are uniformly distributed within a concentric circular region (radius $R_{max} = R_0/2$). The black dotted line in Fig. 5d illustrates the $\gamma$-dependent hyperuniformity parameter $\bar{\Lambda}(\infty)$, averaging over $N_{con} = 400$ local isomorphic configurations for each $\gamma$ point to mitigate finite-size fluctuations. Our results align with ref. 5, showing that $\bar{\Lambda}(\infty)$ tends to increase monotonically with higher $\gamma$ values. At $\gamma = 0$, LIC1 exhibits the global minimum of $\bar{\Lambda}(\infty)$, with the highest level of hyperuniformity. As $\gamma$ increases, hyperuniformity decreases for different LICs.

## The HHG yield VS the hyperuniformity of LICs

While simulating the harmonic yield dependence on the parameter $\gamma$, to balance computational feasibility and accuracy, we focused on six distinct LICs at $\gamma = 0, 0.1, 0.2, 0.3, 0.4$, and 0.5. For each $\gamma$, we averaged $\bar{\Lambda}(\infty)$ over $N_{con} = 10$ random configurations. The result exhibits a monotonic increase and closely agrees with the results obtained by averaging over 400 configurations, as shown by the red dotted line in Fig. 5d. The current was averaged over the 10 configurations to obtain the HHG for each LIC. As shown in Fig. 5e, with increasing $\gamma$, the yields of the 11th to 15th harmonics monotonically decrease, consistent with the hyperuniformity. For the 17th and 19th harmonics, an anomalous enhancement occurs for $\gamma > 0.3$. We speculate that this is due to the presence of Z and ST VEs at $\gamma = 0.3$ in Fig. 5c. These VEs, characterized by the second-highest and highest occurrences of thin rhombi, exhibit high atomic density, leading to some electrons being bound. The bounded electrons will undergo rapid oscillations along the short diagonal of the thin rhombus under the external field, potentially causing an increase in high-frequency harmonic yield. Hence, we conclude that quasicrystal hyperuniformity can be characterized by HHG yields. Hyperuniformity also affects electron transport, notably reflected in current amplitude variations. In our simulation (see Supplementary Note 7 and Supplementary Fig. 15a), increasing the $\gamma$ parameter reduces hyperuniformity, leading to a gradual decline in current amplitudes for distinct LICs. This is also attributed to diverse VE choices, influencing the similarity of electron motion environments and affecting both the coherence and amplitude of current. Special VEs (X, Y, Z, and ST)[4], with higher proportions of thin rhombi and atomic density, modify the electron momentum more strongly. Due to finite size constraints, the proportions of these VEs fluctuate within different configurations of one LIC, which is also reflected in the current amplitudes (see Supplementary Fig. 15b).

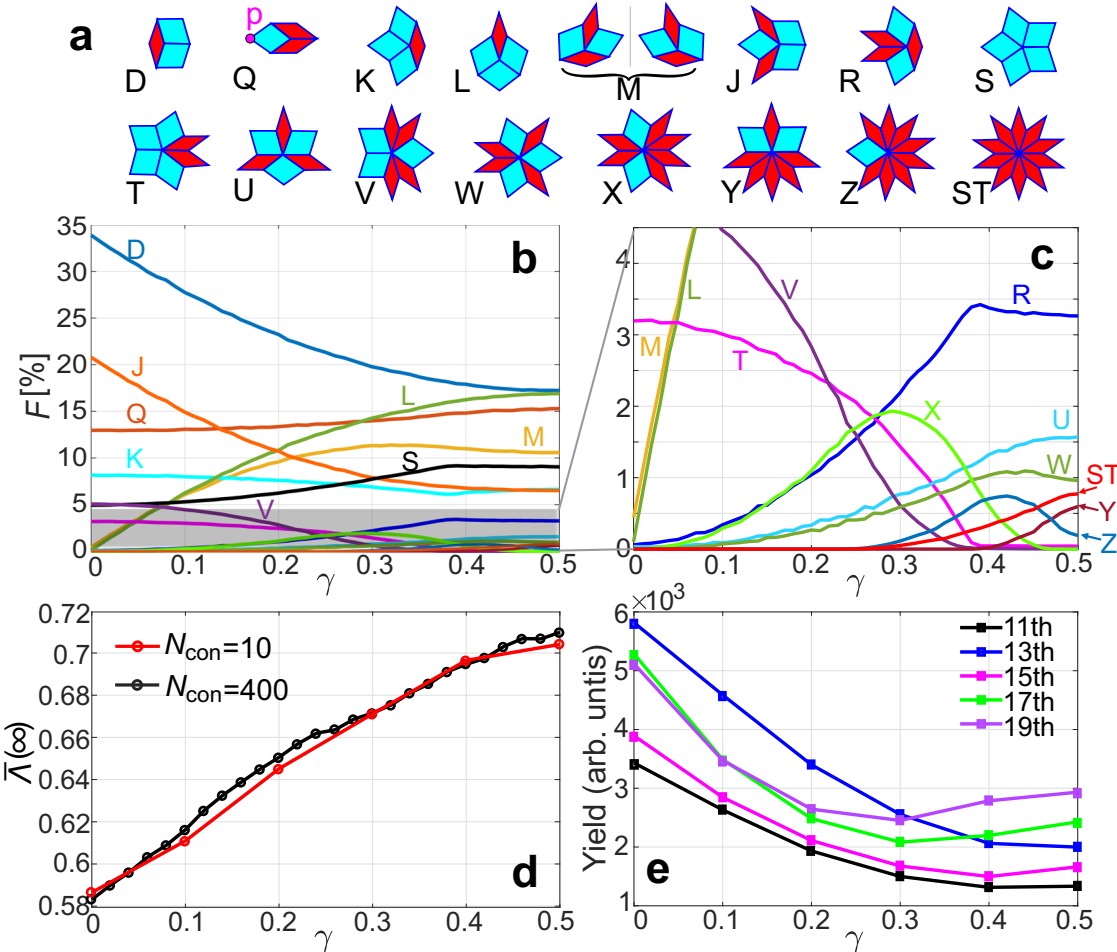

**Fig. 5 | The γ-dependent VEs frequency, superuniformity, and harmonic yield.**
**a** 16 potential VEs in the generalized Penrose quasicrystal. **b** VEs frequency $F$ as a function of $\gamma$. **c** A localized enlargement of the lower shaded region in (**b**). **d** Quasicrystal hyperuniformity parameter $\bar{\Lambda}(\infty)$ as a function of $\gamma$. Black and red

lines represent results considering 400 and 10 local isomorphic configurations at each $\gamma$, respectively. **e** $\gamma$-dependence of 11th to 19th harmonic yields, shown as solid lines in black, blue, magenta, green, and purple, respectively. Laser parameters align with Fig. 2. Source data are provided as a Source Data file.

## Discussion

Distinguishing LICs in quasicrystals is an important task since they have many distinct physical properties. Traditional EDS relies on the scattering near the nuclei, resulting in almost the same pattern for different LICs. HHG comes from the laser-induced electron motions with a wide range in quasicrystals, which encodes more information than EDS. Here we proposed to use high-order harmonic spectroscopy with more freedom to identify different LICs. When a linearly polarized laser illuminates the quasicrystal, the HHG yield of different LICs can have obvious distinctions. In addition, for our quasicrystal models, the orientation-dependent harmonic yield tends to exhibit $C_{10}$ symmetry (forbidden by crystallography), which is consistent with the orientation order and can take the role of the quasicrystal HHG fingerprint. Therefore, different LICs can be distinguished by HHG in yield and orientation-dependence. Furthermore, we use harmonic yields to reveal the quasicrystal hyperuniformity, which is dependent on the VE distribution of various LICs and impacts electron transport properties, such as current amplitude. Our work establishes a link between quasicrystal HHG and LIC, reveals the effect of VE distribution on electron scattering, and provides a new all-optical scheme for studying the structural properties of quasicrystal. It may also be applied to study the topological properties of quasicrystals in the future. Further, we note that the quasicrystal can be approximated by a periodic system with tunable unit cell sizes[4]. This approach will aid in understanding how

periodicity breaking and electron incoherent scattering affect quasicrystal harmonic yield in future studies.

## Methods

### Initial state acquisition

Before the laser field arrives, the electronic structure of the 2D generalized Penrose quasicrystal, including eigenenergies $\varepsilon_n$ and eigenstates $\psi_n$, can be described by the stationary Schrödinger equation,

$$\hat{H}_0 \psi_n = \varepsilon_n \psi_n. \tag{1}$$

Here, $\hat{H}_0$ is the system's stationary Hamiltonian,

$$\hat{H}_0 = \frac{\hbar^2}{2m}\boldsymbol{\nabla}^2 + V_{ion}(\mathbf{r}). \tag{2}$$

Similar to the tight-binding vertex model (considering interactions between atoms at rhombus vertices), the localized Gaussian potentials are adopted to describe the atomic Coulomb potential,

$$V_{ion}(\mathbf{r}) = -\sum_{\nu=1}^{N_{atom}} V_0 \exp\left[-\frac{(\mathbf{r}-\mathbf{r}_\nu)^2}{2\alpha^2}\right], \tag{3}$$

where $V_0 = 1$ a.u., $\alpha = 0.8$ a.u., and $\mathbf{r}_\nu$ is the atomic position determined by the GDM. As shown in Fig. 1, the $N_{atom}$ atoms of the 2D quasicrystal

configurations are distributed within a circle of radius $R_0 = 16l$, where $l = 8$ a.u. is the rhombus edge length. In the simulation, we construct the Hamiltonian matrix $H_0$ in discrete 2D real space using the finite difference method, then diagonalize it to obtain the eigenenergies and eigenstates, applying open boundary conditions. The number of initially occupied valence electrons equals the total number of atoms, $N_{atom}$. As shown in Supplementary Note 1 and Supplementary Figs. 1 and 2, when studying the system's static properties (such as localization and energy spectrum), we use a circular computational box with a radius of $R_0$ to avoid interference from free electron states near the boundary. To simulate harmonic radiation from the laser-quasicrystal interaction, we construct the $H_0$ matrix in a larger rectangular box with side lengths $2R_0 + 2L_{ab}$ and diagonalize it to obtain the $N_{atom}$ initial states (with $\varepsilon_n < 0$). During electron evolution, an absorption function of the form $\cos^{1/8}$, with width $L_{ab} = 20$ a.u., is applied near the box boundary to prevent electron reflection.

## Time-dependent Schrödinger equation

The electron dynamics in the laser-quasicrystal interaction are described by the 2D TDSE in the length gauge,

$$ih\frac{\partial}{\partial t}\psi(\mathbf{r}, t) = \left[\hat{H}_0 + \mathbf{E}(t) \cdot \mathbf{r}\right]\psi(\mathbf{r}, t). \tag{4}$$

$\mathbf{E}(t)$ is the laser electric field. When the angle between the laser principal axis and the $x$-axis is $\theta$ (see Fig. 2b), the components in $x$ and $y$ directions of the electric field can be written as,

$$E_x(t) = E_{x,\theta=0}(t)\cos\theta - E_{y,\theta=0}(t)\sin\theta, \tag{5}$$

$$E_y(t) = E_{x,\theta=0}(t)\sin\theta + E_{y,\theta=0}(t)\cos\theta. \tag{6}$$

Here,

$$E_{x,\theta=0}(t) = E_0/\sqrt{1+\epsilon^2}f(t)\cos(\omega_0 t), \tag{7}$$

$$E_{y,\theta=0}(t) = \epsilon E_0/\sqrt{1+\epsilon^2}f(t)\sin(\omega_0 t). \tag{8}$$

$E_0$, $\epsilon$, $f(t)$, $\omega_0$, and $L_t$, respectively, denote the field strength, ellipticity, envelope, carrier frequency, and total duration. In the calculation, we apply the split-operator method to evolve the $N_{atom}$ initial states $\psi_i$ independently to $\psi_i(\mathbf{r}, t)$. The $\chi$ ($= x$ or $y$) direction harmonic spectrum $H_\chi(\omega)$ is given by the Fourier transform $H_\chi(\omega) = |\mathcal{F}[J_\chi(t)]|^2$. $J_\chi(t)$ is the total laser-induced current in the $\chi$ direction, with

$$J_\chi(t) = \sum_i^{N_{atom}} j_{\chi,i}(t), \tag{9}$$

and $j_{\chi,i}(t) = \langle\psi_i(\mathbf{r}, t)|\hat{p}_\chi|\psi_i(\mathbf{r}, t)\rangle$. The total HHG is $H_{tot}(\omega) = H_x(\omega) + H_y(\omega)$, and the yield $I_n$ of the $n$th harmonic is evaluated by,

$$I_n = \int_{(n-0.5)\omega_0}^{(n+0.5)\omega_0} H_{tot}(\omega)d\omega. \tag{10}$$

In the actual calculation, we set the time step $dt = 0.25$ a.u., and the grid spacing $dx = dy = 0.25$ a.u.

## Data availability

Source data are provided in this paper.

## Code availability

Computer codes of this work are available from the corresponding author upon request.

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

## Acknowledgements

X.B. acknowledges the support from the Hubei Provincial Natural Science Foundation of China (Grant No. 2024AFA029), the National Natural Science Foundation of China (Grant Nos. 12274421 and 12121004), and the CAS Project for Young Scientists in Basic Research (Grant No. YSBR-059). J.L. acknowledges the support from the National Natural Science Foundation of China (Grant No. 12304383) and the China Postdoctoral Science Foundation (Grant Nos. BX20220311 and 2023M733577).

## Author contributions

X.B. designed research; J.L. and X.B. performed research; J.L. and X.B. analyzed data; and J.L. and X.B. wrote the paper.

## Competing interests

The authors declare no competing interests.
