## [Transparent Peer Review file · Nature Communications]

Distinguishing Local Isomorphism Classes in Quasicrystals by High-Order Harmonic Spectroscopy

Corresponding Author: Professor Xue-Bin Bian

Version 0:

Reviewer comments:

Reviewer #1

(Remarks to the Author)

The authors discuss higher-order harmonics from quasicrystals based on theoretical and numerical calculations and show that they can be distinguished in different local isomorphism classes (LICs) as differences in harmonic intensities and their crystal angle dependence (Figs. 2 and 3). This work is expected to be of interest not only to researchers in the field of crystallography, but also in the fields of nonlinear optics and ultrafast spectroscopy, as the new spectroscopic method clarifies differences in states that are difficult to distinguish by the conventional techniques of electron beam diffraction spectroscopy (EDS). Thus, this referee recommends publication in Nature communications if the authors can satisfactorily answer the following questions.

(1) What is the reason for the choice of harmonic orders treated in Figures 2, 3, etc.? Is it a result that cannot be obtained with lower orders such as SHG and THG? What are the advantages of higher-order harmonics?

(2) In the generation of higher harmonics from solids, it is known that in semiconductors and insulators, interband and intraband transitions play an important role in the generation of higher harmonics (Phys. Rev. B 77, 075330(2008), Nat. Photonics 8, 119 (2014), Nat. Phys. 18, 874(2022)). It would be helpful to comment on the relationship between harmonics and these transitions (or currents) in the present quasicrystal. A related question is: How are these physical processes related to the differences in harmonic intensities in the present results that allow us to distinguish LICs and vertex environments (VEs) shown in Figs. 2, 3, and 5?

(3) Could you comment on the universality of the present results, i.e., for what kind of electronic states of materials, such as metals, semiconductors, insulators, etc., are they valid? Also, what kind of electronic state materials are assumed in the present calculations? A prospective discussion should elaborate on the extent to which there is potential for application to other quasicrystals.

Reviewer #2

(Remarks to the Author)

The manuscript authored by Jia-Qi Liu Xue-Bin Bian refers to the problem of distinguishing different tilings belonging to different local isomorphic classes based on analysis of their electron diffraction spectra. The problem is very subtle since tilings of different isomorphic classes produce very similar electron patterns since the long-range order for tilings of different classes is essentially the same. This idea has been, indeed, as authors claim themselves in the manuscript, a long-standing problem in the crystallography of quasiperiodic systems, for which the ambiguity of structure description employing tilings is more profound compared to periodic materials.

The authors introduce a new method of distinguishing the tiling of different local isomorphic classes with the use of high-order harmonic spectroscopy, which gives insight into the local environment of the diffracting electrons in the system, which may be different for structures based on different local isomorphic class. The idea is very interesting and seems to work, as it is supported by the analysis presented in the manuscript. I found the results physically sound and, thus, the paper worth

publishing.

General remarks:

1) Authors use a "real-space" method of generating different classes of Penrose tiling (from early Steinhardt et al. papers), utilizing the pentagrid within the dual method. It is however another gold standard for that using the superspace description. The tiling can be generated by projecting the superspace (5-dimensional) structure via a window (or atomic/occupation domain). By shifting the domain along the z-direction tilings of different local vertices arrangements (vertex environments) can be obtained. It is called in literature a Generalized Penrose tiling. For reference see:

Pavlovitch, A. & Kleman, M. (1987). J. Phys. A Math. Gen. 20, 687–702.

Ishihara, K. N. & Yamamoto, A. (1988). Acta Cryst. A44, 508–516.

M. Chodyn, P. Kuczera, J. Wolny, Acta Cryst. (2015). A71, 161–168

Please, add information on this alternative method in the introductory parts, since it is, probably, better known to the broad audience.

2) According to early papers cited above, the Penrose tiling (LIC 1 in your notation) there are 7 basic VE → including D from Figure 5A of your manuscript. I do not understand why D is excluded from your consideration. D and Q are the basic patches to be considered a phason flip (seen in Figure 4B - see my comments below). Also, the most general Penrose tiling can exhibit one another VE, not mentioned in your manuscript. It is very similar to M from your Figure 5A but mirrored along the short diagonal of the bottom thin rhombus (pertaining thin rhombus is then pointing more to the left). I do not know this additional M-like VE affects your results (probably not at all), but please, consider this problem.

In the list below I give suggestions for changes and improvements to the manuscript:

1) page 4, line 97: "of the our model" → fix

2) on page 4 line 120 abbreviation 'LP' is introduced for 'linear polarization'. The abbreviation appears later in the text only twice in the caption of Figure 2. Perhaps introducing this abbreviation is not needed. Please, reconsider.

3) in the part "HHG of Different LICs" in chapter 3. Results on page 7 line 165 I suggest focusing the readers' attention more on figure 1, to which the phrase 'Configurations 3 and 5 belong...' refers. Maybe add: (see Figure 1)

4) discussion on page 8 lines 203-214 in my opinion refers to a phenomenon known in aperiodic crystals as 'phason flip' → the mismatch between two patterns (moire) in Figure 4B can be resolved by introducing a flip of a single (central) atom in hexagonal patches of tiles: two fat + one thin, or two thin + one fat. The shift of the atom is what we consider just a phason flip. It is known that by introducing phasonic disorder (in terms of phason flips) the continuous transformation between Penrose tiling and Generalized Penrose tiling can be obtained (see papers by C. Henley or H.R. Trebin or more recently I. Buganski). Perhaps you could relate your method (HHG) with the problem of phasonic disorder. Could you, please, answer, if and how HHG could be used to measure the amount of phasonic disorder (understood in terms of phason flipped tile configurations in the system) in the system? Perhaps adding a small paragraph to the manuscript on the subject would be beneficial.

5) equation S1 in suppl. materials: this structure factor is well-defined for vertex-decoration-only structures with a single kind of atom decorating vertices. Please, add a comment.

6) starting with the bottom of page 1 of suppl. mat.: notation with square brackets for denoting alternative decorations and their property looks in some places like citations for reference positions. Perhaps change it.

7) caption of Figure S4 in suppl. mat: 'belong' → 'belonging' and please highlight/distinguish bold for 'x' direction.

Reviewer #3

(Remarks to the Author)

Please see attached DOC file.

Version 1:

Reviewer comments:

Reviewer #1

(Remarks to the Author)

Although the authors have partially answered the question, the results of the present calculations only show differences between Configuration 1 and Configuration 3, which still raises questions about the significance of this study.

In particular, the following points are not clear and I think that the overall significance of this study has not reached the general level of Nature Communications.

1) It is good that the crystal angle dependence of configuration 1 and configuration3 is different, but it is unclear what mechanism is responsible for this crystal angle dependence or it is not clear what is the origin of the difference.

2) Configuration3-5 shows similar, crystal angle dependence, and it is not clear how those differences can be revealed or the detailed difference can be inferred from the HHGs.

Then, the fact that the proposal is based on computational results only, and it is unclear whether the differences in configuration can be measured in principle, and thus I think that a journal such as Communications Physics or other related journals would be appropriate.

Reviewer #2

(Remarks to the Author)

I accept the response to my review from the authors. In my opinion, the article can be published in the current form.

Reviewer #3

(Remarks to the Author)

The authors have addressed all my criticism. I recommend publication.

The authors thank all the reviewers very much for the valuable comments and for finding that our results are “interesting”, “valuable”, and “beautiful”. The original reports are enclosed in black color, our responses are in blue. The changes made in the manuscript are marked in blue color and listed below in green color.

Response to Reviewer #1

The authors discuss higher-order harmonics from quasicrystals based on theoretical and numerical calculations and show that they can be distinguished in different local isomorphism classes (LICs) as differences in harmonic intensities and their crystal angle dependence (Figs. 2 and 3). This work is expected to be of interest not only to researchers in the field of crystallography, but also in the fields of nonlinear optics and ultrafast spectroscopy, as the new spectroscopic method clarifies differences in states that are difficult to distinguish by the conventional techniques of electron beam diffraction spectroscopy (EDS). Thus, this referee recommends publication in Nature communications if the authors can satisfactorily answer the following questions.

Response: Thank you very much for your efforts in reviewing our work as it helps a lot to improve our manuscript significantly. Below is our reply to each comment.

(1) What is the reason for the choice of harmonic orders treated in Figures 2, 3, etc.? Is it a result that cannot be obtained with lower orders such as SHG and THG? What are the advantages of higher-order harmonics?

Response: Thank you for the nice question. Lower-order harmonics correspond to the perturbative regime. As shown in Fig. R1, the THGs of different LICs are quite close. The harmonics in the plateau region correspond to the nonperturbative regime. They are closely linked to the electron motion with a bigger excursion distance, encoding more information about atomic arrangements. Thus they can distinguish different LICs efficiently. As a result, our study focuses on the significant differences in harmonic yields between LICs in the plateau region (11th to 21st orders) of Fig. R1.

Fig. R1. Comparison of HHG between configurations 1 and 3.

Here, we also studied the orientation dependence of lower and higher-order harmonics. As shown in Fig. R2(a), the orientation-dependent third-order harmonic yields of configurations 2 (LIC1) and 4 (LIC2) are nearly identical, making it difficult to distinguish between LICs. This issue doesn't occur for higher-order harmonics in the plateau region. As shown in Fig. R2(b), the orientation-dependent 11th harmonics are clearly separated into two groups corresponding to different LICs, with similar behavior seen in other harmonic orders of the plateau region (see Fig. R9).

Fig. R2. Comparison of orientation-dependent 3rd and 11th order harmonic yields for different configurations (LIC1: configurations 1, 2; LIC2: configurations 3, 4, 5).

Change: We interpret why we don't choose the lower-order harmonics to distinguish different LICs in the revised version.

(2) In the generation of higher harmonics from solids, it is known that in semiconductors and insulators, interband and intraband transitions play an important role in the generation of higher harmonics (Phys. Rev. B 77, 075330(2008), Nat. Photonics 8, 119 (2014), Nat. Phys. 18, 874(2022)). It would be helpful to comment on the relationship between harmonics and these transitions (or currents) in the present quasicrystal. A related question is: How are these physical processes related to the differences in harmonic intensities in the present results that allow us to distinguish LICs and vertex environments (VEs) shown in Figs. 2, 3, and 5?

Response: We thank you for this very important comment. The results presented in our work are the total contributions, which are hard to separate into inter- or intra-band contributions in k space technically because the periodic property is lost in quasicrystals. However, we can do this in the energy space [see Guan et al., Phys. Rev. A 93, 033852 (2016)]. For example, we take the i th valence state ϕ_i^v as the initial state to solve TDSE $\psi_i(t)$. The interband current involving transitions between different eigenstates (the index is n) in different bands (the index is $b = v$ or c) can be written as,

$$j_{inter}^i = -\frac{e}{m} \sum_{b \neq b'} \sum_{nn'} \langle \psi_i(t) | \phi_n^b \rangle \langle \phi_n^b | \hat{p} | \phi_{n'}^{b'} \rangle \langle \phi_{n'}^{b'} | \psi_i(t) \rangle.$$

The intraband current involves transitions between states in the same band and can be written as,

$$j_{intra}^i = -\frac{e}{m} \sum_b \sum_{nn'} \langle \psi_i(t) | \phi_n^b \rangle \langle \phi_n^b | \hat{p} | \phi_{n'}^b \rangle \langle \phi_{n'}^b | \psi_i(t) \rangle.$$

Thus, the total interband and intraband currents involving N_e valence electrons are given by $J_{inter} = \sum_i^{N_e} j_{inter}^i$ and $J_{intra} = \sum_i^{N_e} j_{intra}^i$, respectively.

Fig. R3. The variation of high harmonic spectra for (a) configuration 1 and (b) configuration 4 with the number of atoms in the system. The laser parameters are consistent with Fig. 2 in the main text. Dashed lines are used as visual guides to more intuitively illustrate the higher yields of 11th to 19th order harmonics in LIC1 compared to LIC2.

Even in a 1D system, computing the projection $\langle \psi_i(t) | \phi_n^b \rangle$ of the time-dependent wave function onto a series of eigenstates $\{\phi_n^b\}$ and distinguishing between intraband and interband currents is computationally intensive due to the large number of integrals involved. In our manuscript, simulating HHG in quasicrystals involves solving the 2D TDSE N_e times independently. The increased dimensionality makes it exceptionally difficult to distinguish the intraband and interband currents for each valence electron. To ensure computational feasibility, we have to reduce the system size. We first tested the convergence of HHG in the current quasicrystal model for the number of atoms. For configurations 1 and 4, Fig. R3 demonstrates the convergence of high harmonic spectra with the system's size under linearly polarized driving lasers, reaching near convergence when the number of atoms $N = N_e = 456$. Therefore, in the following simulations of intraband and interband HHG, our model includes 456 atoms instead of the 991 atoms mentioned in the manuscript.

As shown in Supplementary Figs. 1(a)~1(e) of the Supplementary Information, we sorted the eigenenergies and divided the spectrum into the valence band (VB, below zero) and conduction band (CB, above zero). From Supplementary Figs. 1(a)~1(e), the minimum band gaps (MBGs) between the VB and CB are $0.173 a.u.$ ($\approx 10\omega$) for configuration 1 and $0.203 a.u.$ ($\approx 11\omega$) for configuration 4, with ω corresponding to the laser frequency at $\lambda = 2.6 \mu m$. Figures R4(a) and R4(b) show the intraband and interband HHG components for configurations 1 and 4. Comparing the relative intensities of intraband and interband HHGs shows that harmonics in the plateau region, with energies above MBG, arise from interband transitions, while lower-order harmonics involve contributions from both mechanisms. Comparing the inter+intra results with standard TDSE, we find differences in both low and high-frequency regions. This discrepancy arises from truncating higher-energy CB states for computational feasibility, which reduces both intraband and higher-order interband harmonic yields. Thus, interband polarization dominates the

differences in harmonic yields among LICs, influenced by VE-modulated eigenstate localization. Similar to crystals, we believe that in quasicrystals, interband harmonic corresponds to electron-atom scattering under an external field in real space. When the electron travels a sufficiently long distance, its coherence is affected by scattering from various VEs, leading to differences in harmonic yield across different LICs.

Fig. R4. Contribution of inter- and intraband HHGs in (a) configuration 1 and (b) configuration 4, with $N = N_e = 456$. Intra, inter, and intra+inter HHGs are shown by the green solid line, red dashed line, and black solid line, respectively. The laser parameters are consistent with Fig. 2 and Fig. R3. For comparison, HHG from the standard TDSE calculation is highlighted in gray. Dashed magenta line is also used as visual guides to more intuitively illustrate the higher yields of 11th to 19th order harmonics in LIC1 compared to LIC2.

Change: We have added Figs. R3 and R4 to the Supplementary Information to show the contribution of inter- and intra-band transitions. We have cited the mentioned papers in the reference. We have added some sentences in the main manuscript to interpret that the main differences in HHG spectra between different LICs come from the interband transitions, in which the long excursion distance with more scattering by VEs allows different LICs to be distinguished.

(3) Could you comment on the universality of the present results, i.e., for what kind of electronic states of materials, such as metals, semiconductors, insulators, etc., are they valid? Also, what kind of electronic state materials are assumed in the present calculations? A prospective discussion should elaborate on the extent to which there is potential for application to other quasicrystals.

Response: We thank you for this question. In solid-state physics, materials are classified based on their electronic band structure into metals (partially filled bands) and non-metals (fully filled VBs). Non-metals are further divided into semiconductors (band gap of 0~2 eV) and insulators (gap > 2 eV). As shown in Supplementary Figs. 1(a)~1(e) of the Supplementary Information, our simulation models with fully occupied VBs and energy gaps of 0.1730~0.2030 *a.u.* ($\approx 4.71\sim 5.52\text{eV}$). Our model corresponds to a typical insulator in terms of band gap. In ultrafast processes, semiconductors, and insulators, both gapped systems differ mainly in electron excitation probability and laser damage threshold, while their harmonic generation mechanisms are similar. We, therefore, suggest that even if the band gap is reduced below 2 eV, suitable laser wavelengths and field strengths would still allow for the distinction of different LICs, making our

model applicable to other gapped systems.

Change: We added a prospective discussion on the universality of the present results in other gapped quasicrystals.

Response to Reviewer #2

The manuscript authored by Jia-Qi Liu Xue-Bin Bian refers to the problem of distinguishing different tilings belonging to different local isomorphic classes based on analysis of their electron diffraction spectra. The problem is very subtle since tilings of different isomorphic classes produce very similar electron patterns since the long-range order for tilings of different classes is essentially the same. This idea has been, indeed, as authors claim themselves in the manuscript, a long-standing problem in the crystallography of quasiperiodic systems, for which the ambiguity of structure description employing tilings is more profound compared to periodic materials.

The authors introduce a new method of distinguishing the tiling of different local isomorphic classes with the use of high-order harmonic spectroscopy, which gives insight into the local environment of the diffracting electrons in the system, which may be different for structures based on different local isomorphic class. The idea is very interesting and seems to work, as it is supported by the analysis presented in the manuscript. I found the results physically sound and, thus, the paper worth publishing.

Response: We appreciate your very positive comments which are very encouraging.

General remarks:

1) Authors use a "real-space" method of generating different classes of Penrose tiling (from early Steinhardt et al. papers), utilizing the pentagrid within the dual method. It is however another gold standard for that using the superspace description. The tiling can be generated by projecting the superspace (5-dimensional) structure via a window (or atomic/occupation domain). By shifting the domain along the z-direction tilings of different local vertices arrangements (vertex environments) can be obtained. It is called in literature a Generalized Penrose tiling. For reference see:

Pavlovitch, A. & Kleman, M. (1987). *J. Phys. A Math. Gen.* 20, 687–702.

Ishihara, K. N. & Yamamoto, A. (1988). *Acta Cryst.* A44, 508–516.

M. Chodyn, P. Kuczera, J. Wolny, *Acta Cryst.* (2015). A71, 161–168

Please, add information on this alternative method in the introductory parts, since it is, probably, better known to the broad audience.

Response: Thank you for the valuable suggestion and recommended references. We agree with you and apologize for not discussing other methods for generating 2D quasicrystals. As you pointed out, a proper introduction to these 2D quasicrystal construction methods may broaden readers' interest and provide diverse structural generation techniques for future quasicrystal harmonic studies. The earliest and most well-known 2D quasicrystal structure is the Penrose tiling, which can be generated using the "matching rules", "self-similarity transformation", "generalized dual method", or "superspace projection". Beyond Penrose tiling, the generalized dual method or superspace projection of a 5D crystal (as noted by the reviewer) can also produce other quasiperiodic tilings, which cannot be described by matching rules and do not belong to the Penrose LIC.

Change: In the revised manuscript, we have referred to all 2D quasicrystals from the generalized

dual method as "generalized Penrose quasicrystals", replacing "Penrose-like quasicrystals" to align with established terminology. We have expanded the introduction to include the methods for generating 2D (generalized) Penrose quasicrystals and added citations to the relevant literature.

2) According to early papers cited above, the Penrose tiling (LIC 1 in your notation) there are 7 basic VE -> including D from Figure 5A of your manuscript. I do not understand why D is excluded from your consideration. D and Q are the basic patches to be considered a phason flip (seen in Figure 4B - see my comments below). Also, the most general Penrose tiling can exhibit one another VE, not mentioned in your manuscript. It is very similar to M from your Figure 5A but mirrored along the short diagonal of the bottom thin rhombus (pertaining thin rhombus is then pointing more to the left). I do not know this additional M-like VE affects your results (probably not at all), but please, consider this problem.

Response: Thank you for your questions. (1) Regarding vertex environment (VE) D: As noted, phason flip, introducing phasonic disorder, can be easily applied to Q and D [J. Appl. Cryst. (2020). 53, 904-913]. In our manuscript, we did not separately analyze the impact of specific VEs on harmonic spectra or exclude their contributions. We attribute differences in harmonic yields among LICs to the disruption of electron coherence caused by VE complexity. HHG in generalized Penrose quasicrystals results from electron scattering across various VEs under an external laser field. In other words, we considered both Q and D's contributions to the harmonics. In our manuscript, we discuss how vertex p in Q forms various VEs across different LICs to illustrate how VE diversity affects harmonics. This analysis is also applicable to other VEs, including D.

(2) Regarding M-like VE, we thank you for the careful review. Figure 5a shows 16 vertex environments, consistent with Refs. [4, 5]. We agree with you. Besides the 16 vertex environments listed, there is an additional M-like VE that cannot be derived from the M VE through rotation or translation within the 2D plane. In our manuscript, we classify both M-like VE and M VE as the same VE, M, defined by two fat and two thin rhombuses. By the local isomorphism requirement, M-like VE and M VE always appear in pairs within a configuration. For example, both M-like VE and M VE are found in configuration 3 (LIC2), as shown in Fig. R5. Thus, in our simulations, both M-like VE and M VE simultaneously influence the harmonics.

Fig. R5. The distributions of M and M-like vertex environments in configuration 3, indicated by blue and yellow points, respectively.

Change: In the revised manuscript, we emphasized the contributions of all VEs as follows: "In fact, all VEs in the system are interconnected through complex electron scattering and collectively contribute to harmonic radiation."; To ensure a complete classification of VEs, we revised the definition of M to encompass both M and M-like VEs and updated Fig. 5a accordingly.

In the list below I give suggestions for changes and improvements to the manuscript:

1) page 4, line 97: "of the our model" -> fix

Response: We are sorry about that.

Change: We have deleted "the" in the revised manuscript.

2) on page 4 line 120 abbreviation 'LP' is introduced for 'linear polarization'. The abbreviation appears later in the text only twice in the caption of Figure 2. Perhaps introducing this abbreviation is not needed. Please, reconsider.

Response: We agree with you.

Change: In the revised manuscript, we have removed the abbreviation 'LP'.

3) in the part "HHG of Different LICs" in chapter 3. Results on page 7 line 165 I suggest focusing the readers' attention more on figure 1, to which the phrase 'Configurations 3 and 5 belong...' refers. Maybe add: (see Figure 1)

Response: Thank you for the nice suggestion.

Change: We have added the relevant reading guidance in the revised manuscript.

4) discussion on page 8 lines 203-214 in my opinion refers to a phenomenon known in aperiodic crystals as 'phason flip' -> the mismatch between two patterns (moire) in Figure 4B can be resolved by introducing a flip of a single (central) atom in hexagonal patches of tiles: two fat + one thin, or two thin + one fat. The shift of the atom is what we consider just a phason flip. It is known that by introducing phasonic disorder (in terms of phason flips) the continuous transformation between Penrose tiling and Generalized Penrose tiling can be obtained (see papers by C. Henley or H.R. Trebin or more recently I. Buganski). Perhaps you could relate your method (HHG) with the problem of phasonic disorder. Could you, please, answer, if and how HHG could be used to measure the amount of phasonic disorder (understood in terms of phason flipped tile configurations in the system) in the system? Perhaps adding a small paragraph to the manuscript on the subject would be beneficial.

Response: Thank you for the insightful suggestion, which has brought our attention to the issue of phason flips in quasicrystals. This is a new topic for us. As described by Bugański et al. in their recent work [J. Appl. Cryst. 53, 904 (2022)]: "In the case of the decagonal quasicrystal, there are two simplest possible ways of reshuffling the rhombuses: the flip of a patch of two fat and one thin rhombus (fft) or two thin and one fat rhombus (ftt)." Thus, phason flips can be easily performed for Q and D, as shown in Fig. R6.

Fig. R6. Two possible flips: fft (VE D's position O to O') and ft (VE Q's position P to P').

We agree with the reviewer. The mismatch between the two patterns in Fig. 4b indeed corresponds to a phason flip for Q or D. However, phason flips between locally isomorphic configurations affect only specific patches, not all Q and D. This differs from the random phason flips introduced for each patch in [J. Appl. Cryst. 53, 904 (2022)], which results in tiling disorder. According to your suggestion and following the approach of Bugański et al. in [J. Appl. Cryst. 53, 904 (2022)], we start with a Penrose tiling (configuration 1, LIC1) and introduce phason disorder by randomly flipping Q and D. Figures R7(a)~R7(c) show the configuration with no flips, after 5 cycles of flips, and after 1000 cycles of flips, respectively. As the number of flip cycles increases, the rhombus pattern in the tiling becomes more random, in line with the literature [J. Appl. Cryst. 53, 904 (2022)].

Fig. R7. (a)~(c) The evolution of the tiling of configuration 1 after randomization with an increasing number of phason flip iterations (from left to right). (d) shows the harmonic spectra of configuration 1 after different flip cycles, compared with configuration 3 (LIC2). (e)~(g) show the statistics of the VE type and count for (a)~(c), respectively.

In Fig. R7(d), the phasonic disorder affects the harmonic yield in the plateau region. In configuration 1, after 5 flip cycles, the yields of the 15th, 17th, and 19th-order harmonics decrease and approach those of configuration 3 (LIC2). After 1000 flip cycles, the previously reduced

harmonics are enhanced again, and their yields are comparable to those without phason flips. We also attribute these changes to the diversity and distribution of VEs. As shown in Fig. R7(e), without flips, VEs are mainly D and J, with fewer others. Thus, scattering is dominated by D and J, resulting in less coherence loss and higher harmonic yield. After 5 cycles of flip, vertex environments become more evenly distributed among Q, M, K, L, and J, see Fig. R7(f). The scattering of diverse VEs reduces electron coherence and lowers the harmonic yield. As iteration reaches 1000, in Fig. R7(g), the VEs predominantly are D and M, with fewer of the others. Consequently, electron coherence loss diminishes, and the harmonic yield increases. In addition, we can reasonably speculate that in the 2D cases, phason disorder may change the orientation dependence of the HHG signal.

Change: We have added the related papers on the phason flips to the references. The discussion about this question has been added to the revised manuscript and Supplementary Information to explore the potential impact of phasonic disorder on HHG.

5) equation S1 in suppl. materials: this structure factor is well-defined for vertex-decoration-only structures with a single kind of atom decorating vertices. Please, add a comment.

Response: Thank you for your suggestion. We agree with your point. Equation S1 (now Eq. (4) in the Supplementary Information) describes the electron diffraction pattern (or structure factor) of a 2D quasicrystal decorated at vertices with a single type of atom. As you noted, if each vertex (or rhombus) was decorated with a more complex atomic structure or multiple types of atoms, the equation would need to be adjusted accordingly. This would affect the details of the electron diffraction pattern and involve a more complex mathematical description [J. Appl. Cryst. 53, 904 (2022)].

Change: In the revised manuscript and Supplementary Information, we emphasize that each vertex in models is decorated with a single identical atom.

6) starting with the bottom of page 1 of suppl. mat.: notation with square brackets for denoting alternative decorations and their property looks in some places like citations for reference positions. Perhaps change it.

Response: Thank you for pointing out the issue with our use of square brackets.

Change: In the revised version, we have replaced the square brackets with parentheses for Supplementary Information.

7) caption of Figure S4 in suppl. mat: 'belong'  'belonging' and please highlight/distinguish bold for 'x' direction.

Response: Thank you for pointing out these writing issues.

Change: We have corrected them in the revised version.

Response to Reviewer #3

Liu and Bian report on a beautiful theory work on high-harmonic emission from various quasicrystals with different configurations belonging to two distinct Local Isomorphism Classes (LICs). They observe that configurations belonging to the same LIC emit very similar high-harmonic spectra, whereas different LICs emit spectra with slightly different powers. Furthermore, they report that the orientation dependence of two exemplary configurations

belonging to two different LICs is significantly different, albeit both exhibit the expected 10-fold symmetry (given the definition of 5 axis for the quasicrystals).

They explain their findings in terms of the different average vertex environment of the various LICs and configurations. Configurations in the same LIC exhibit similar vertex environments or, in the words of the authors, "a finite atomic arrangement centered around one vertex is found consistently in another configuration". Thus, electron trajectories responsible for high-harmonic emission launched from any vertex will experience, on average, the same scattering of other configurations belonging to the same LIC. On the other hand, the vertex environment, and thus the scattering, of a different LIC is different – thus explaining why different LICs yield different harmonic spectra.

The authors corroborate this finding by exploring high-harmonic emission with a continuously-varying γ parameter, which changes the “hyperuniformity” of the quasicrystal or, in other words, the representation of the 16 possible vertex environments. They find that a more uniform distribution of possible vertex environments leads to decreased high-harmonic emission, in agreement with the previous findings: more VEs mean more inhomogeneous system, thus more scattering.

I find the results are mostly well justified, the science sound, the manuscript is well written and relatively easy to parse. Exploring the role of “quasi-symmetry” on high-harmonic generation pushes a frontier of the field, where previously only perfectly symmetric crystals have been considered. In particular, I find the authors’ insights into the role of non-perfect symmetry to be very relevant and potentially useful to further our understanding of “standard” high-harmonic emission from real-life crystals, where disorder is always present to some extent. For example: room-temperature crystals where phonons cause a deviation of the position of the atoms from perfect periodicity (with the electron-hole trajectories “freezing in” the phonon); glasses and ceramics, where long-range order is lost; poly-crystals, where grain boundaries cause scattering and loss of coherent. To my judgement, all these aspects have been largely neglected, yet they are important as they expand the reach of attosecond science to more complex systems. Thus, I recommend publication of this work in Nature Communications. However, I do have some comments that I would like the authors to address before publications:

Response: We thank you very much for these highly positive comments and for emphasizing the potential values of our research.

1. I find the relevance of the work poorly argued for. Why should we care about high-harmonic spectroscopy of quasi-crystals? The authors do comment on 30deg twisted bilayer graphene, but is this the only relevance? The authors should make a stronger case for their work, possibly connecting to existing literature on high-harmonic from solids (see above for potential arguments).

Response: Thank you very much for this comment. Besides the disorder and the phonon effect pointed out by the reviewer, the first motivation behind this manuscript was to search for a novel and efficient condensed-matter HHG source. As you mentioned above, most of the publications on solid HHG are related to perfect crystals. However, the maximum photon energy in crystal HHG is still very low. Recently, HHGs have also been observed in liquid thin film [15,16], indicating

that periodicity is not a prerequisite for harmonic generation. Thus, we predict that quasicrystals, with their unique order between crystals and liquids, provide a new possible way to produce higher HHG photons.

Our second motivation is to investigate the relationship between forbidden rotational symmetries and orientation-dependent harmonics in quasicrystals. Here, we focus on structurally simple 2D quasicrystals in our theoretical study. Experimentally, various 2D quasicrystals have been synthesized, making the detection of quasicrystal harmonics feasible. For instance, 2D quasicrystals with 8-fold (e.g., Cr-Ni-Si alloys [Phys. Rev. Lett. 59, 1010 (1987)]), 10-fold (e.g., AlMn, AlFe alloys [Phys. Rev. Lett. 55, 1461 (1985)]), and 12-fold (e.g., Ni-Cr alloys [Phys. Rev. Lett. 55, 511 (1985)]) symmetries have been observed. These quasicrystals exhibit periodic atomic structures along one direction and quasiperiodic arrangements in the perpendicular plane. Additionally, as mentioned in the manuscript, twisted 2D materials (e.g., graphene) offer further experimental avenues for obtaining 2D quasicrystals.

Motivation 3: HHGs capture ultrafast electron dynamics, which are linked to the system's atomic arrangement and electronic structure. The complex atomic structures of higher-dimensional (2D or 3D) quasicrystals limit the study of their electronic properties. HHG offers a potential all-optical method to explore both their atomic structures and electronic properties. Moreover, quasicrystals are inherently linked to topological systems [Phys. Rev. Lett. 109, 106402 (2012), Phys. Rev. B 91, 064201 (2015), Phys. Rev. Lett. 103, 013901 (2009), Phys. Rev. Lett. 121, 126401 (2018)], making HHG a promising tool for exploring quasicrystals' topological properties.

Change: We borrowed the reviewer's comments to underscore the significance of real materials (often exhibiting disorder and non-perfect symmetry) for advancing attosecond science, and emphasized the relevance of our quasicrystal system for understanding harmonic generation in complex systems. Additionally, we noted the importance of quasicrystal HHG for developing novel high-energy photon sources, identifying crystallographic forbidden symmetries, and probing quasicrystal structures and electronic properties.

2. In the abstract, the authors explain that high-harmonic spectroscopy offers time resolution. However, this aspect is not argued for any further in the manuscript. What would be a useful time-dependent problem to address in quasi-crystals?

Response: Thank you for pointing this issue out. HHG is a subcycle process. It could provide time resolution in principle. However, we didn't investigate it further since the main topic of this paper is about distinguishing LICs. We are sorry for writing it in the abstract but without further study. In future work, the pump-probe scheme may give us some useful information about the time-dependent electron-hole evolution with higher time resolution.

Change: We delete the "time resolution" from the abstract.

3. The authors use a 2.6 μ m laser wavelength and an intensity of 3×10^{13} W/cm². Why? Typically, one chooses a wavelength that is much smaller than the bandgap of the material, and an intensity below the damage threshold. Please comment.

Response: You are right. The laser parameters chosen in our simulation are to avoid the damage of quasicrystals. Lower photon energy in the driving laser means a smaller absorbing section, which allows a higher laser intensity to produce a nonperturbative effect without damaging the target. Similar laser parameters can be found in the literature. For example, in the Ref. [Diam.

Relat. Mater. 82, 165 (2018)], the authors studied the harmonic radiation of bulk diamond, with an energy gap close to our current model, driven by an 800 nm laser with $I_0 = 3 \times 10^{13} \text{W/cm}^2$. Longer wavelengths and short pulse duration are used in our manuscripts to further avoid material damage.

Another consideration to choose laser parameters is to capture the impact of vertex environment diversity in different LICs by HHG. The electron's excursion distance x_0 in the laser field should be sufficiently large. This distance x_0 can be qualitatively estimated as $x_0 \propto \frac{E_0}{\omega^2}$. By increasing the field strength or wavelength, the electron's excursion distance can be extended. For example, in Fig. R8, the differences in harmonic yield between configurations 1 and 3 are small at shorter wavelengths. As the wavelength increases, the differences in the plateau region become more pronounced. Thus, ensuring a sufficiently large electron excursion range is the basis for our choice of laser parameters (intensity and wavelength) in the manuscript.

Fig. R8. The HHG, driven by the linear polarization laser, of configurations 1 and 3 for different wavelength λ , with $I_0 = 3 \times 10^{13} \text{W/cm}^2$. More results can be found in Supplementary Fig. 5 (c) of the Supplementary Information.

Change: We added a discussion on the reason for the choice of laser parameters in the amended paper.

4. Fig. 3: The authors should prove that the orientation dependence of configurations belonging to the same LIC (i.e. configurations 1 and 2) are similar and distinct from those belonging to LIC2 (3-5). Otherwise the authors cannot claim that the orientation dependence can distinguish LICs (it might just distinguish configurations).

Response: We thank the referee for the nice comment. As shown in Fig. R9, we compare the harmonic yields of different orders in the plateau region (11th to 21st order harmonics) with various configurations to demonstrate that orientation-dependent HHG is similar in the same LIC but distinct in different LICs.

In Fig. R9, the orientation-dependent HHG of all configurations is grouped into two clusters based on LICs, as highlighted by the light yellow (LIC1, including configurations 1 and 2) and light gray (LIC2, including configurations 3~5) regions. Configurations within the same LIC show

very similar HHG, while significant difference is observed between different LICs. This supports our earlier conclusion that HHG orientation dependence can distinguish between different LICs.

Fig. R9. Comparison of orientation-dependent 11th~21th order harmonic yields for different configurations (LIC1: configurations 1, 2; LIC2: configurations 3, 4, 5).

Change: We have expanded this discussion in Supplementary Note 4 and Supplementary Fig. 10 of the revised Supplementary Information.

5. At line 187, the authors state that “the electron trajectories corresponding to different harmonics orders differ, resulting in different θ dependence as a function of harmonics photon energy”. The strength of the harmonics is not necessarily given by the trajectory of the electrons, rather it’s *mostly* given by the ionization rate - which depends on relative orientations of the lattice vertex and laser polarization. What is the role of ionization?

Response: We agree with you. The ionization rate from the valence band to the conduction band is also orientation-dependent. It can be understood by the Keldysh theory. After ionization, the electron-hole trajectory is also orientation-dependent because they may encounter different lattice vertices. We think both effects contribute a lot.

Change: We added the role of the θ -dependent ionization rate from the valence to the conduction bands in the sentences.

6. At line 205, the authors state that “the emitted harmonic radiation exhibits enhanced coherence”. I wouldn’t call it “enhanced coherence”, rather it’s those electrons experiencing different vertex patterns that exhibit loss of coherence.

Response: We agree with you.

Change: In the revised manuscript, we rewrite the sentence as: “If electrons from different

vertices observe different vertex patterns, the emitted harmonic radiations will exhibit suppressed coherence”.

7. Further to #6 above, given the high degree of disorder in this quasi-crystals, I imagine loss of coherence is an important aspect, so the emission process must be quite inefficient. Can the authors comment on this? Is there a way to tune the system to make it progressively more periodic?

Response: Thank you for raising this point. Quasicrystals are not disordered in our minds. It is not periodic, but it obeys a strict rule. It can also be understood by the projection of a higher dimensional crystal on the reduced dimensions. This is the reason why the bright sharp peaks appear in the electron diffraction spectra of quasicrystals. The high degree of disorder will lead to a blurry emission pattern. The types and distributions of VEs in quasicrystals will affect the coherence of harmonic radiation. Thus, coherence loss during electron motion is crucial for distinguishing different LICs. Our manuscript demonstrates that quasicrystals, which exist between crystals and amorphous materials, have the potential to serve as novel sources of high-order harmonics. However, we do not directly compare the HHG efficiency of quasicrystals with that of crystals, as we believe such comparisons are highly challenging. This is due to the many factors influencing solid harmonic yields, mainly divided into laser conditions (e.g., intensity, wavelength, duration) and electronic properties (e.g., energy spectra, localization). The distinct electronic properties of quasicrystals and crystals require different laser conditions for high harmonic efficiency, making the comparison difficult.

We understand the reviewer's concern that the loss of translational periodicity in quasicrystals can lead to incoherent scattering of electrons, which may reduce the efficiency of harmonic generation compared to crystals. To address this issue, one would need to compare quasicrystals and crystals with similar energy spectra, which aligns with the reviewer's suggestion to explore ways to enhance periodicity in the current system. This is theoretically feasible. We have previously explored similar issues in our work on 1D Fibonacci quasicrystal HHG [Phys. Rev. Lett. 127, 213901 (2021)]. There, a generalized Fibonacci model [Phys. Rev. B 41, 10398 (1990)] is used to naturally connect crystals and quasicrystals, yielding similar low-energy harmonic spectra. In the Fibonacci quasicrystal, incoherent electron scattering opens more excitation channels than in crystals, enhancing the high-energy harmonic radiation.

Next, we show how to adjust the current generalized Penrose quasicrystal to enhance its periodicity using the periodic approximants method. By adjusting the star vectors in the generalized dual method with F_n and F_{n+1} (where F_n is the Fibonacci number of the n th generation), Lin et al. [Phys. Rev. Lett. 120, 247401 (2018)] achieved periodic approximants for generalized Penrose quasicrystals. Quasicrystals are approximated by periodic systems with finite-sized unit cells. As n increases, the number of atoms in the unit cell grows, converging towards a perfect quasicrystal. This process is illustrated more intuitively in Fig. R10, which is borrowed from Ref. [Phys. Rev. Lett. 120, 247401 (2018)]. Thus, we could start with a spatial region A . As n increases, the number of unit cells in A decreases, weakening the periodicity. For sufficiently large n , the configuration in A approximates a quasicrystal. Comparing harmonics with varying n will allow us to assess the impact of incoherent scattering, which will be addressed in future work.

Fig. R10. Examples of periodic approximants from two different LI classes (top to bottom) and from three different degrees of approximant (left to right), borrowed from Ref. [Phys. Rev. Lett. 120, 247401 (2018)]. The LIC1 is $\gamma = 0$ (top row) and LIC2 is $\gamma = 0.5$ (bottom row). The number of points in the unit cell for each approximant is $N = 76$ (left column), $N = 199$ (middle column), and $N = 521$ (right column).

Change: In the discussion part of the revised manuscript, we added the discussion of the periodic approximants method to approximate the generalized Penrose quasicrystals, which will help to reveal the effect of incoherent scattering of electrons in quasicrystals on the harmonic yield.

8. Line 227: "... which reduces the HHG coherence and yield." The authors should point to Fig. 5E here.

Response: Yes, it is.

Change: In the revised manuscript, we added a few words "(as we will see in Fig. 5e)" to help readers to understand it.

9. Lines 258-259: regarding the enhancement of HH17-19 for $\gamma > 0.3$. The authors state that this is due to the presence of Z and ST VEs. This is a speculation. The reason given in the next sentences doesn't amount to proof. Can the authors more objectively prove the connection between various VEs and individual harmonics?

Response: We agree that this is a speculation. We cannot objectively prove the connection between various VEs and individual harmonics. The discussion in the manuscript is based on the comparison of Figs. 5c and 5e. $\gamma = 0.3$ corresponds to the appearance of Z and ST VEs in Fig. 5c. It also corresponds to the yield increasing in Fig. 5e. So we guess that there may be some connection between them. However, it is beyond our ability to directly separate the contribution of each VE in each harmonic.

Change: We changed the tone to be as soft as possible to compare Figs. 5c and 5e, the connection is just a possible qualitative analysis.

The authors thank all the reviewers very much again for your valuable comments and suggestions which helped us a lot to improve our manuscript. We have revised our manuscript accordingly and hope all the referees are satisfied with our responses.

The authors thank all the reviewers very much for your time in the second round of review. The original reports are enclosed in black color, our responses are in blue. The changes made in the manuscript are listed below in green color.

Response to Reviewer #1

Although the authors have partially answered the question, the results of the present calculations only show differences between Configuration 1 and Configuration 3, which still raises questions about the significance of this study.

In particular, the following points are not clear and I think that the overall significance of this study has not reached the general level of Nature Communications.

Response: Thank you very much for your efforts in reviewing our work again. The obvious differences in HHG spectra from Configurations 1 and 3 are strong evidence that high-order harmonic spectroscopy can distinguish LICs in quasicrystals. As you wrote in the first round review report, it is “difficult to distinguish by the conventional techniques of electron beam diffraction spectroscopy (EDS)”. It was also pointed out by Reviewer #2 that this is “a long-standing problem in the crystallography of quasiperiodic systems”. So we think that this study solves a critical problem in this research field, and the significance should reach the general level of Nature Communications.

1) It is good that the crystal angle dependence of configuration 1 and configuration 3 is different, but it is unclear what mechanism is responsible for this crystal angle dependence or it is not clear what is the origin of the difference.

Response: Thank you for the comment. The different angle dependence of configurations 1 and 3 is shown in the section “HHG of Different LICs”, while the mechanism and the origin are discussed in the following sections “Real-space Electron Motion Perspective” and “VEs and Hyperuniformity”. Here, we summarize it as follows: (1) Like crystals, quasicrystals with long-range order are anisotropic, producing sharp bright spots in EDS. This differs from disordered systems like glass or liquids, where isotropic atomic arrangements yield diffuse diffraction patterns. Consequently, quasicrystal harmonic emission exhibits crystal-like angular dependence and features crystallographically forbidden C_{10} symmetry, reflecting the orientational order of the quasicrystal and aligning with the symmetry of its EDS. The orientation order of the quasicrystal is governed by the star vectors in Fig. 1(a), which determine the alignment of each rhombus; specifically, each rhombus edge aligns parallel to a star vector. (2) Next, we address the origin of differences in harmonic emission between distinct local isomorphism classes (LICs). Our manuscript attributes these differences to variations in the types and distributions of vertex environments (VEs). In a laser field, electron motion in quasicrystals is spatially confined in a finite region, and the HHG comes from scattering between electrons and atoms. Increased VE diversity creates greater atomic pattern variability in the finite region of electron motion, weakening electron coherence and reducing harmonic yield. As shown in Fig. 5(b), the types and numbers of VEs differ across different LICs, allowing us to differentiate LICs by comparing harmonic intensities. In the quasicrystal, the spatial distribution and orientation of each VE (in Fig. 5(a)) are also controlled by the orientation order and the quasiperiodic order (modulated by γ), leading to distinct angular dependencies in HHG across different LICs.

Change: In the revised manuscript, we add a sentence at the end of the section “HHG of Different LICs” to tell readers that the mechanism and origin will be discussed in detail in the next two sections.

2) Configuration 3-5 shows similar, crystal angle dependence, and it is not clear how those differences can be revealed or the detailed difference can be inferred from the HHGs.

Response: Thank you for this comment. In Figs. 2(c), 2(d), and Supplementary Figs. 4–7 and 10, we compare the harmonics of different configurations within the same LIC, finding almost identical spectra. This result agrees with our expectation and it guarantees that the harmonic radiation is independent of illumination position. In the manuscript, we explain this result using the definition of LICs: for two locally isomorphic configurations, any finite-sized local pattern in one configuration can be found in the other. This means any finite local electron motion environment is shared by both locally isomorphic configurations, yielding identical harmonic spectra. The configurations 3~5 mentioned by the reviewer all belong to LIC2 in our manuscript, so we expect their harmonic spectra to be nearly identical, including similar angular dependencies in harmonic yield (Supplementary Fig. 10). We did not use HHG to distinguish different configurations within the same LIC. As indicated in the title of this paper, our goal is to distinguish different LICs.

Change: In the revised manuscript, we emphasize that "different configurations within the same LIC should exhibit nearly identical harmonic emission."

Then, the fact that the proposal is based on computational results only, and it is unclear whether the differences in configuration can be measured in principle, and thus I think that a journal such as Communications Physics or other related journals would be appropriate.

Response: Although this work is based on computational methods, the results are accurate and logical. We also discussed that the laser condition and the candidate quasicrystal system, such as twisted bilayer graphene and alloy system, are accessible now. We believe that the differences can be measured in principle. The intersection of quasicrystals and attosecond physics is still nascent, we believe that our manuscript will lay the groundwork for future research. The high impact of Nature Communications will accelerate the progress in this research direction.

Response to Reviewer #2

I accept the response to my review from the authors. In my opinion, the article can be published in the current form.

Response: We appreciate your efforts in reviewing and thank you very much for the recommendation.

Response to Reviewer #3

The authors have addressed all my criticism. I recommend publication.

Response: We appreciate your help in improving this manuscript and thank you very much for the recommendation.

The authors thank all the reviewers very much again for the comments and recommendations. We have revised our manuscript accordingly and hope all the referees are satisfied with our responses.

Liu and Bian report on a beautiful theory work on high-harmonic emission from various quasicrystals with different configurations belonging to two distinct Local Isomorphism Classes (LICs). They observe that configurations belonging to the same LIC emit very similar high-harmonic spectra, whereas different LICs emit spectra with slightly different powers. Furthermore, they report that the orientation dependence of two exemplary configurations belonging to two different LICs is significantly different, albeit both exhibit the expected 10-fold symmetry (given the definition of 5 axis for the quasicrystals).

They explain their findings in terms of the different average vertex environment of the various LICs and configurations. Configurations in the same LIC exhibit similar vertex environments or, in the words of the authors, "a finite atomic arrangement centered around one vertex is found consistently in another configuration". Thus, electron trajectories responsible for high-harmonic emission launched from any vertex will experience, on average, the same scattering of other configurations belonging to the same LIC. On the other hand, the vertex environment, and thus the scattering, of a different LIC is different – thus explaining why different LICs yield different harmonic spectra.

The authors corroborate this finding by exploring high-harmonic emission with a continuously-varying γ parameter, which changes the "hyperuniformity" of the quasicrystal or, in other words, the representation of the 16 possible vertex environments. They find that a more uniform distribution of possible vertex environments leads to decreased high-harmonic emission, in agreement with the previous findings: more VEs mean more inhomogeneous system, thus more scattering.

I find the results are mostly well justified, the science sound, the manuscript is well written and relatively easy to parse. Exploring the role of "quasi-symmetry" on high-harmonic generation pushes a frontier of the field, where previously only perfectly symmetric crystals have been considered. In particular, I find the authors' insights into the role of non-perfect symmetry to be very relevant and potentially useful to further our understanding of "standard" high-harmonic emission from real-life crystals, where disorder is always present to some extent. For example: room-temperature crystals where phonons cause a deviation of the position of the atoms from perfect periodicity (with the electron-hole trajectories "freezing in" the phonon); glasses and ceramics, where long-range order is lost; poly-crystals, where grain boundaries cause scattering and loss of coherent. To my judgement, all these aspects have been largely neglected, yet they are important as they expand the reach of attosecond science to more complex systems. Thus, I recommend publication of this work in Nature Communications. However, I do have some comments that I would like the authors to address before publications:

1. I find the relevance of the work poorly argued for. Why should we care about high-harmonic spectroscopy of quasi-crystals? The authors do comment on 30deg twisted bilayer graphene, but is this the only relevance? The authors should make a stronger case for their work, possibly connecting to existing literature on high-harmonic from solids (see above for potential arguments).
2. In the abstract, the authors explain that high-harmonic spectroscopy offers time resolution. However, this aspect is not argued for any further in the manuscript. What would be a useful time-dependent problem to address in quasi-crystals?
3. The authors use a 2.6um laser wavelength and an intensity of 3×10^{13} W/cm². Why? Typically, one chooses a wavelength that is much smaller than the bandgap of the material, and an intensity below the damage threshold. Please comment.
4. Fig. 3: The authors should prove that the orientation dependence of configurations belonging to the same LIC (i.e. configurations 1 and 2) are similar and distinct from those belonging to LIC2 (3-5). Otherwise the authors cannot claim that the orientation dependence can distinguish LICs (it might just distinguish configurations).
5. At line 187, the authors state that “the electron trajectories corresponding to different harmonics orders differ, resulting in different /theta dependence as a function of harmonics photon energy”. The strength of the harmonics is not necessarily given by the trajectory of the electrons, rather it's *mostly* given by the ionization rate - which depends on relative orientations of the lattice vertex and laser polarization. What is the role of ionization?
6. At line 205, the authors state that “the emitted harmonic radiation exhibits enhanced coherence”. I wouldn't call it “enhanced coherence”, rather it's those electrons experiencing different vertex patterns that exhibit loss of coherence.
7. Further to #6 above, given the high degree of disorder in this quasi-crystals, I imagine loss of coherence is an important aspect, so the emission process must be quite inefficient. Can the authors comment on this? Is there a way to tune the system to make it progressively more periodic?
8. Line 227: “... which reduces the HHG coherence and yield.” The authors should point to Fig. 5E here.
9. Lines 258-259: regarding the enhancement of HH17-19 for $\gamma > 0.3$. The authors state that this is due to the presence of Z and ST VEs. This is a speculation. The reason given in the next sentences doesn't amount to proof. Can the authors more objectively prove the connection between various VEs and individual harmonics?
- 10.